# Learning from Noisy Demonstration Sets via Meta-Learned Suitability Assessor

## Abstract

A noisy and diverse demonstration set may hinder the performances of an agent aiming to acquire certain skills via imitation learning. However, state-of-the-art imitation learning algorithms often assume the optimality of the given demonstration set. In this paper, we address such optimal assumption by learning only from the most suitable demonstrations in a given set. Suitability of a demonstration is estimated by whether imitating it produce desirable outcomes for achieving the goals of the tasks. For more efficient demonstration suitability assessments, the learning agent should be capable of imitating a demonstration as quick as possible, which shares similar spirit with fast adaptation in the meta-learning regime. Our framework, thus built on top of Model-Agnostic Meta-Learning, evaluates how desirable the imitated outcomes are, after adaptation to each demonstration in the set. The resulting assessments hence enable us to select suitable demonstration subsets for acquiring better imitated skills. The videos related to our experiments are available at https://sites.google.com/view/deepdj.

## 1 Introduction

Imagine that you intend to learn how to make a free throw in basketball, which requires you to throw the ball into the basket from a fixed position. Without the proper knowledge, one may observe professional players perform a free throw on YouTube by obtaining numerous exemplary videos. However, learning from every demonstration videos might lead to worse performance, as they may contain unsuitable or even irrelevant content.

The challenge of learning from noisy demonstration sets is as well crucial in the robot imitation learning regime, as demonstrations which are not aligned with achieving the intended goal deteriorate the learning process. The assumptions of optimality (or at least sub-optimality) of the demonstrations are often made in state-of-the-art imitation learning algorithms (Ross & Bagnell, 2010; Ross et al., 2011; Ho & Ermon, 2016; Sermanet et al., 2018). As a result, they are vulnerable to demonstrations that are potentially detrimental to the learning outcomes in the given set. To address assumptions of optimality, in this paper, we aim for a generic framework capable of learning from a noisy demonstration set, via evaluating the suitability of imitated skills judged by task specific heuristics.

Prior works have handled deteriorated imitated outcomes due to noisy demonstration sets by utilizing expected Q-values provided in the demonstrations to avoid learning from bad demonstrated actions (Kim et al., 2013; Nair et al., 2017; Gao et al., 2018). However, these works require demonstrations to have a rich representation such as incorporating the aforementioned Q-values, which may neither be available in other cases nor directly applicable to the agent training environment. Our framework, on the other hand, does not require demonstrations to contain specific information, and hence is able to cope with any forms of expert demonstrations. To be specific, we propose to first assess which demonstrations in the given set might be more suitable *by learning from them*, and then train the agent imitating only a selected subset.

In order to achieve selectively learning from suitable demonstrations, we examine if the learning outcomes are favorable after imitating each demonstration in a set. Typically in robot learning regime, we are able to receive designed feedbacks in the target training environment to evaluate how well the agent performs. Thus, in each task, we predefine specific heuristics for assessing if the agent exhibits imitation learning outcomes desirable for reaching the goals of the tasks.

The key challenges for the feasibility of assessing learned outcomes from each demonstration are the efficiency and generalization ability. A framework should be capable of producing these assessable outcomes as quick as possible, and generalizing to unseen demonstrations. To this end, we propose a framework with the demonstration suitability assessor leveraging meta-learning, where we train adaptive parameters via meta-imitation-learning. The meta-imitation-learned parameters can thus: (1) produce assessable imitated outcomes at testing time quicker than both imitation learning from scratch and fine-tune a pretrained initialization (Finn et al., 2017a), and (2) adapt to imitating newly sampled unseen demonstrations. Overall, the imitated outcomes after adaptation will be judged by task heuristics to indicate the suitability of certain demonstrations. We then train agents using imitation learning from the selected suitable demonstration subsets for obtaining better policies. We demonstrate two empirical approaches to utilize the resulting suitability assessments. One composes new subsets of demonstrations from the top ranked, the other iteratively fine-tunes the meta-learned parameters by strengthening or weakening certain demonstrations in a set according to current suitability judgments, producing a selected suitable subset at convergence.

In some cases, the distribution of demonstration sets can be imbalanced or multi-modal. To prevent over-fitting to certain subsets of such demonstration distributions, we augment the meta-imitation-training with a regularization objective-maximization of mutual information between the demonstration and the induced behavioral differences from imitating it. This additional regularization term aims to make the meta-trained parameters more responsive to the demonstrations being imitated, and as a result, help differentiate better the adapted behaviors from a noisy and diverse set.

We test our framework on four different simulation sports environments in MuJuCo. Our results, both qualitative and quantitative, show that the proposed method outperforms various baselines, including vanilla MAML and fine-tuning a pretrained initialization, on learning better policies from noisy demonstration sets.

## 2 RELATED WORKS

**Imitation Learning and Learning Diverse Behaviors** In imitation learning, an agent tries to acquire skills by learning from demonstrated expert behaviors (Schaal, 1997; Argall et al., 2009; Ziebart et al., 2008; Kuniyoshi, 1989; Hussein et al., 2017; Ross et al., 2011). Ho & Ermon (2016) discover that imitation learning can be formulated as an adversarial training (Goodfellow et al., 2014) between demonstrations and imitated behaviors of the agent (GAIL), which is adopted in this work as the main imitation learning framework. Extension works of GAIL (Wang et al., 2017; Hausman et al., 2017; Li et al., 2017) augment the original algorithm with a latent code or embedding for imitating a diverse set of demonstrations, increasing diversity of the imitated outcomes. Duan et al. (2017) learn an encoder-decoder framework for digesting multiple demonstrations and achieving one-shot imitation learning during testing time. Hester et al. (2018) train a deep q-network by initializing the replay buffer with expert demonstrations. The demonstration assessing process of our work can also be viewed as leveraging the results of few-shot imitation learning from diverse demonstrations. However, the emphasis of our goal is centered on utilizing the assessments to acquire better skills learning from selected suitable demonstrations.

**Learning from Imperfect Demonstrations** Despite remarkable performances in imitative learning of complex control behaviors from prior works, most of the algorithms assume the optimality (or at least sub-optimality) of the demonstrations. The given demonstration set is usually presumed to contain only *helpful* demonstrations: it is not noisy. Kaiser et al. (1995) presents several types of experts violating the optimality assumption and basic ways to cope with them. Tomasello (2016) provides psychological evidences that infants imitate rationally, and that they are more prone to certain demonstrations that do not only fundamentally make sense but effectively help the development of their skills. Grollman & Billard (2011) utilize failure demonstrations as negative constraint on the exploration to prevent learning agent from approaching behaviors likewise. Kim et al. (2013) casts learning from limited demonstrations as a relaxed large-margin constrained optimization problem. Several recent works (Gao et al., 2018; Nair et al., 2017) also assume the expert representation contains expected Q-values and utilize them to filter bad demonstrated actions. However, these additional information may not be available in other cases such as learning from raw video demonstrations (Sermanet et al., 2016; 2018). Moreover, specifically for Q-values utilized in aforementioned works, the assumption of these information being applicable to the agent training

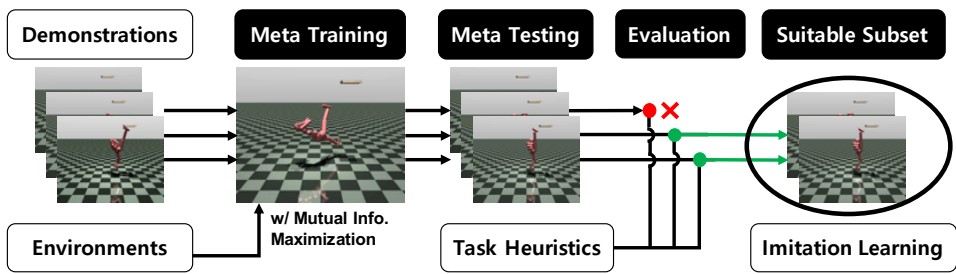

Figure 1: **Illustration of the framework: We first meta-imitation train a set of adaptive parameters on the entire demonstration set. And then, during meta-testing phase, the imitation learning outcomes exhibited after adapting to a particular demonstration are assessed with predefined task heuristics. The assessment result leads to a selected suitable subset for the agent to learn from.**

environment does not necessarily hold. We thus aim for a more general framework capable of coping with any forms of demonstrations, while this work mainly based off state-action pairs.

**Meta Learning in Imitation Learning** Our work is also closely related to a series of recent advancements in meta learning, that we as well seek fast adaptive parameters which should learn how to learn from certain demonstrations. Finn et al. (2017a) propose a gradient-based model-agnostic framework (MAML) for training a policy prior that can quickly adapt to newly sampled tasks during testing (termed as meta-testing) time by small number of gradient update steps. In another work (Finn et al., 2017b) MAML is adopted to train a set of adaptive parameters achieving gradient-based one-shot imitation learning during meta-testing time. As an extension of MAML with regularization in an inner imitative learning objective, we are, to our best knowledge, the first to utilize the meta learning framework to evaluate a demonstration set for learning a better policy from selected demonstrations.

## 3 PRELIMINARIES AND PROBLEM SETUP

A finite-horizon Markov Decision Process (MDP) can be defined as $\mathcal{M} = (S, A, P, R, \gamma, p_0, T)$, where $S$ and $A$ denotes state and action spaces, $P : S \times A \times S \to \mathbb{R}_+$ denotes the state transition probability, $R : S \times A \to \mathbb{R}$ is a reward function, $p_0$ an initial state distribution, $\gamma$ is the reward discount factor, and $T$ is the horizon of the MDP. Let $\tau = (s_0, a_0, ..., s_T, a_T)$ be a trajectory of state and action pairs, the associated reward can be written as $R(\tau) = \sum_{t=0}^{T} \gamma^t R(s_t, a_t)$. Reinforcement learning seeks a policy $\pi_\theta(a|s)$ parameterized by $\theta$ that maximizes the expected total return: $\mathbb{E}_{a_t \sim \pi_\theta, s_t \sim S}[R(\tau)]$. In imitation learning regime, demonstrated expert behaviors are given while the reward function is missing, and the agent acquire desired skills learning from these demonstrations.

We now formulate our problem setup: given a noisy demonstration set $V = \{v_1, ..., v_n\}$, where each $v_i \in V$ can be represented as state-action pairs: $\{(s_{i,t}, a_{i,t})\}$, and several environment variants $E = \{e_1, ..., e_k\}$, where each $e_i$ denotes different goal in a multi-goal learning task. We require a suitability assessment score $S_{v_i}|_{e_j} \in \mathbb{R}$ of a demonstration $v_i$ under $e_j$ for judging and selecting suitable demonstrations. And hence, the assessment scores $\{S_{v_i}|_{e_j}\}$ can be the reference for selecting the most suitable demonstration as $\arg\max_i S_{v_i}|_{e_j}, v_i \in V$, or a subset $\{v_{opt}\}$ for imitation learning.

## 4 METHOD

Our goal is to learn a good policy from a noisy demonstration set by imitating the selected suitable demonstrations. In order to produce imitated outcomes efficiently for suitability evaluation, we utilize the adaptation process in meta-learning framework, Model-Agnostic Meta-Learning (MAML) (Finn et al., 2017a), to meta-imitation-learn a set of fast adaptive parameters from the given demonstration set. At meta-testing time, the parameters should adapt to each sampled demonstration to produce assessable learning outcomes, leading to selected suitable subsets for the agent to learn from. To alleviate insensitivity imitating a particular demonstration to an unbalanced set distribution, motivated by (Chen et al., 2016), we regularize the inner imitation objective of MAML with maximization of mutual information between the demonstration and the induced behavioral differences from imitating it. Fig.1 depicts the overview of our framework, and the pseudo codes are provided in the appendix.

### 4.1 META-TRAINING FOR FAST ADAPTIVE PARAMETERS

Following the conventions as described in MAML (Finn et al., 2017a;b), meta-imitation-learning has two objectives, where the inner objective tries to optimize for adapting to a sampled individual demonstration, while the outer objective (meta objective) tries to optimize for fast adaptation across multiple newly sampled demonstrations. Denote $f_\theta$ as the policy model mapping observations to actions parameterized by $\theta$, the desired adaptive meta-parameters. The inner objective in the MAML-based meta-imitation-learning is an imitation objective, where $\theta$ is updated to $\theta_i^{'}$ after learning from demonstration $v_i$ by gradient descent with learning rate $\alpha$, which can be written as:

$$\theta_i^{'} = \theta - \alpha \nabla_\theta \mathcal{L}_{v_i}(f_\theta), \ v_i \in V \tag{1}$$

Here $\mathcal{L}_{v_i}$ denotes the imitation loss imitating the sampled demonstration $v_i$. And hence the meta objective and the corresponding meta-imitation-update can be derived as (with meta learning rate $\beta$):

$$\min_\theta \sum_{v_i \in V} \mathcal{L}_{v_i}(f_{\theta_i^{'}}), \text{ where } \theta \leftarrow \theta - \beta \nabla_\theta \sum_{v_i \in V} \mathcal{L}_{v_i}(f_{\theta_i^{'}}) \tag{2}$$

#### 4.1.1 INNER IMITATION LEARNING UPDATE

The inner imitative update in Eq.1 is not limited to certain methods, we hence adopt two major imitation learning algorithms: generative adversarial imitation learning and behavioral cloning.

**Generative Adversarial Imitation Learning (GAIL):** Our primary adopted imitation learning method (Ho & Ermon, 2016). For each demonstration $v_i \in V$, we maintain a discriminator $D_i$ dedicated to treating demonstration $v_i$ as real data in the GAIL setting contributing to $\mathcal{L}_{v_i}$ in Eq. 1. Specifically, each inner update from $\theta$ to $\theta_i^{'}$ is computed with reward $-\log(D_i(s, a))$ via TRPO (Schulman et al., 2015) update rule. In this paper, we denote this reward as $R_{IL}$ and the sparse task reward as $R_{task}$. In a standard imitation learning scheme, both $R_{IL}, R_{task}$ should be taken into account. However, $R_{task}$ would bias the training of the adaptive parameters $\theta$, that $\theta$ should be optimized for quick adaptation to any given demonstration rather than succeeding the task. Therefore, we drop the $R_{task}$ term and denote our reward definition as $R(s_t, a_t) = R_{IL}, 0 \le t \le T$.

In order to adapt to unseen demonstrations, we train a *meta-discriminator* $D_{meta}$ along with training the adaptive policy parameters $\theta$. Similar to Eq.2, $D_{meta}$ is meta-updated as:

$$D_{meta} \leftarrow D_{meta} - \beta \sum_i [\hat{\mathbb{E}}_{\tau_i}[\nabla log(D_{meta,i}^{'}(s, a))] + \hat{\mathbb{E}}_{v_i}[\nabla log(1 - D_{meta,i}^{'}(s, a))]], \tag{3}$$

where $\tau_i$ denotes the trajectories sampled from $f_{\theta_i^{'}}$ (after learning from demonstration $v_i$), $v_i$ is the trajectory of demonstration as one episode of state-action pairs (expert trajectory in GAIL setting). Each $D_{meta,i}$ is the updated discriminator parameters during the inner update for demonstration $v_i$, which follows the GAIL update rule for discriminator.

**Behavioral Cloning:** In behavioral cloning, the inner update in Eq.1 is simply a supervised training, where $\mathcal{L}_{v_i} = \frac{1}{T} \sum_t \|f_\theta(s_{i,t}) - \hat{a_{i,t}}\|_2$ using a mean squared error with $\hat{a_{i,t}}$ denoting the expert action from demonstration $v_i$ at time-step $t$.

### 4.2 MUTUAL INFORMATION REGULARIZER

In order to overcome the potential challenge of learning from an unbalanced demonstration distribution, we augment the inner update in meta-training phase with a regularization loss: maximization of the mutual information $\mathcal{I}(v_i, \Delta\pi_{v_i})$ between the induced behavioral differences and the demonstration being imitated. $\Delta\pi_{v_i}$ here denotes the differences of behaviors induced after an inner update learns from a particular demonstration $v_i$, which we estimate using a sample-based method. Maximizing $\mathcal{I}(v_i, \Delta\pi_{v_i})$ can be approximated by optimizing the following variational lower bound:

$$\mathcal{I}(v_i, \Delta\pi_{v_i}) \ge \mathbb{E}[logQ(v_i|\Delta\pi_{v_i})] + H(v_i) = R_{MI}(\pi_{v_i}, Q), \tag{4}$$

where $Q$ denotes the posterior network jointly trained with the adaptive policy parameters. Since $v_i$ is drawn from a fixed given demonstration distribution (a set), $H(v_i)$ can be treated as a constant, we thus only need to consider optimizing $Q$. $Q$ can be trained by estimating $\Delta\pi_{v_i}$ through sampling

trajectories before and after the inner imitation update, denoted as $\tau_{old}$ and $\tau_{new}$. We implement $Q$ using a siamese LSTM network taking as inputs both the old and new trajectories. The output of the posterior network is an encoded feature of the demonstration $v_i$. We sample two sets of trajectories for both $\tau_{old}$ and $\tau_{new}$ as one used for training $Q$, and the other (test set) for estimating $\mathbb{E}[logQ(v_i|\Delta\pi_{v_i})]$ to be used as the regularization reward. We then augment the original reward function in GAIL with $R_{MI} = -\mathbb{E}[logQ(v_i|\Delta\pi_{v_i})]$ as follows:

$$R(s_t, a_t) = \begin{cases} R_{IL} + R_{MI} & \text{if } t = 0 \\ R_{IL} & \text{if } 0 < t \le T \end{cases}, \quad R_{MI} = -\mathbb{E}[logQ(v_i|\Delta\pi_{v_i})] \tag{5}$$

However, this regularization term should be applied to the parameters before the imitation update, since it is interpreted as the probability of such behavioral differences induced by a particular demonstration $v_i$. Denote the parameters for inner imitation update from demonstration $v_i$ as $\theta_i$, our inner objective thus becomes:

$$\mathcal{L}_{v_i}(f_{\theta_i}) = -\mathbb{E}_{\tau_{old} \sim f_{\theta_i}, v_i}[\sum_{t=0}^{T} R_{v_i}(s_{t,old}, a_{t,old})]$$

$$R_{v_i} = R_{MI_i} + R_{IL_i} \tag{6}$$

$$\Rightarrow \mathcal{L}_{v_i}(f_{\theta_i}) = \mathcal{L}_{MI_i}(f_{\theta_i}) + \mathcal{L}_{IL_i}(f_{\theta_i})$$

and hence the update where we revert $\theta_i'$ back to $\theta_i$ with the additional regularization term can be written as:

$$\theta_i - \alpha\nabla_{\theta_i}\mathcal{L}_{v_i,old}(f_{\theta_i})$$

$$= \theta_i - \alpha\nabla_{\theta_i}\mathcal{L}_{IL_i,old}(f_{\theta_i}) - \alpha\nabla_{\theta_i}\mathcal{L}_{MI_i,old}(f_{\theta_i}) \tag{7}$$

$$= \theta_i' - \alpha\nabla_{\theta_i}\mathcal{L}_{MI_i,old}(f_{\theta_i})$$

Which implies we can compute the gradients with respect to the old trajectories but update on the new parameters $\theta_i'$. In practice, we implement an additional value network dedicated for this additional regularization reward. As for behavioral cloning, we update the policy parameters $\theta$ utilizing policy gradients computed from this $R_{MI}$, while other parts remained as standard supervised training.

### 4.3 META-TESTING: JUDGING

During meta-testing, we evaluate each demonstration $v_i \in V$ and select the most suitable one(s) to imitate judged by some task heuristics $\mathcal{K}$ under environment variant $e_j$. Each adaptation at meta-testing time will be run for a few iterations. $\mathcal{K}$ is essentially a scoring function which takes as input trajectories generated from a policy $\pi$ and outputs a real-valued number, ie. $\text{score}_\pi = \mathcal{K}(\tau_\pi) \in \mathbb{R}$. Denote $\pi_\theta$ as the meta-trained policy, and $\pi_{\theta_i}$ is the policy adapted to demonstration $v_i$, the demonstration suitability assessment score is then computed as:

$$S_{v_i}|_{e_j} = \mathcal{K}(\tau_{\pi_{\theta_i}}|_{e_j}) - \mathcal{K}(\tau_{\pi_\theta}|_{e_j}) \tag{8}$$

The most suitable demonstration $v_{i_{opt}|e_j}$ is then selected by:

$$i_{opt} = \arg\max_i S_{v_i}|_{e_j} = \arg\max_i \mathcal{K}(\tau_{\pi_{\theta_i}}|_{e_j}) - \mathcal{K}(\tau_{\pi_\theta}|_{e_j}), \quad \forall v_i \in V \tag{9}$$

### 4.4 APPROXIMATE OPTIMAL SUBSET SELECTION

To select a subset of suitable demonstrations to learn from, we design two empirical approaches that utilize assessment scores obtained in the meta-testing phase. One approach, which is further described in Sec. 5.2 as one of our evaluation metrics, takes the top-$K$ ranked demonstrations as the selected subset. The other approach is an iterative algorithm that searches for a suitable subset using a slightly adjusted version of our meta-imitation-learning framework. The intuition for this approach is described as follows: we introduce a weight $c_i$, initialized as 1.0, for each $v_i$ in the demonstration set. During meta-update, the meta-prior $\theta$ is updated by a *weighted sum* of gradients computed when adapting to each demonstration $v_i$, ie. $\theta \leftarrow \theta - \beta\sum_{v_i \in V} c_i \cdot \nabla_\theta\mathcal{L}_{v_i}(f_{\theta_i'})$. After each meta-testing, $c_i$ is adjusted based on the difference between current assessment score $S_{v_i}|_{e_j}$ and the baseline score obtained from the meta-parameters $S_\theta|_{e_j}$. At convergence, we select a subset $\{v_i|c_i > \epsilon\}$, where $\epsilon$ is a predefined threshold. We include the pseudo code for obtaining such subset in the appendix B.

## 5 EXPERIMENTS

The goals of our experiments are to: (1) Compare different methods on assessing and ranking a given demonstration set according to the suitability of each demonstration. (2) Examine if our meta-learning-based frameworks can generalize such suitability assessment to unseen demonstrations. (3) Provide empirical approaches to learn a better policy from noisy demonstration set. For brevity, we will term the demonstration suitability assessments as DSA. We evaluate the effectiveness of the following methods on learning from a noisy demonstration set in four experimental environments:

- **Avg Fine-Tune:** A framework that fine-tunes towards a particular demonstration with a set of parameters pre-trained using the entire demonstration set.
- **MAML:** The standard MAML framework as described in (Finn et al., 2017a).
- **MAML + MI:** A MAML framework regularized with mutual information maximization of the demonstration and the induced behavioral differences, our core method as described in Sec. 4.2

### 5.1 ENVIRONMENTS AND SETUPS

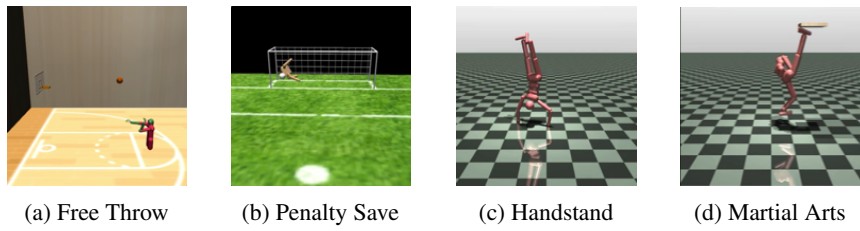

| (a) Free Throw | (b) Penalty Save | (c) Handstand | (d) Martial Arts |

Figure 2: Sample snapshots of the four sports simulation environments in MuJuCo

We built four different simulation sports environments as illustrated in Fig.2. These environments aim to resemble real sports activities and their task heuristics are intuitively designed to fit human preferences. We sample generated state-action pairs $\{(s_t, a_t)\}$ from RL pre-trained agents during different stages of their RL training to compose the demonstration set. Details in the appendix A.

- **Free Throw:** As shown in Fig.2a, the thrower needs to make a basketball free-throw from a fixed position, where it scores if the ball is successfully thrown into the basket. Two datasets are tested containing 8 and 20 demonstrations respectively. The latter one with 20 demonstrations includes demonstrations from a taller thrower agent for verifying our concept that successful demonstrations are not necessarily the suitable ones, it has to depend on the learning agent.
- **Penalty Save:** As shown in Fig.2b, an ant agent is required to jump at the right timing with the proper force to block an automated incoming penalty shot. This is a multi-goal task since the incoming shots can have different directions and speed. There are 12 demonstrations in this dataset.
- **Handstand:** As shown in Fig.2c, a humanoid is required to withstand an upside-down handstand position. The appropriate body-pose is defined as: the hands must touch the ground while feet are raised over a threshold height. There are 15 demonstrations in this dataset.
- **Martial Arts**: As shown in Fig.2d, a humanoid is required to make a jump kick to kick a target placed higher than the height of the humanoid. There are 10 demonstrations in this dataset.

### 5.2 EVALUATION METRICS

Here we explain the metrics used for evaluating and comparing our method and the baselines:

- **Top-$K$ combinations:** After obtaining an empirical rank of the given set sorted by the adapted heuristics scores, we can combine the top-$K$ ranked demonstrations to compose some potentially better datasets. Eg. top-3 means such dataset consists of 3 demonstrations ranked as the top 3 in the set according to the heuristics scores generated during adaptation using one of the methods. The empirical rank generated by our method and the baselines are thus compared by the performances of agents learning from different top-$K$ demonstration sets using imitation learning from scratch.
- **Adapted Behaviors Distance (ABD):** We compare the state-action pairs generated from the adapted parameters and the optimal behaviors (experts) by computing the following euclidean

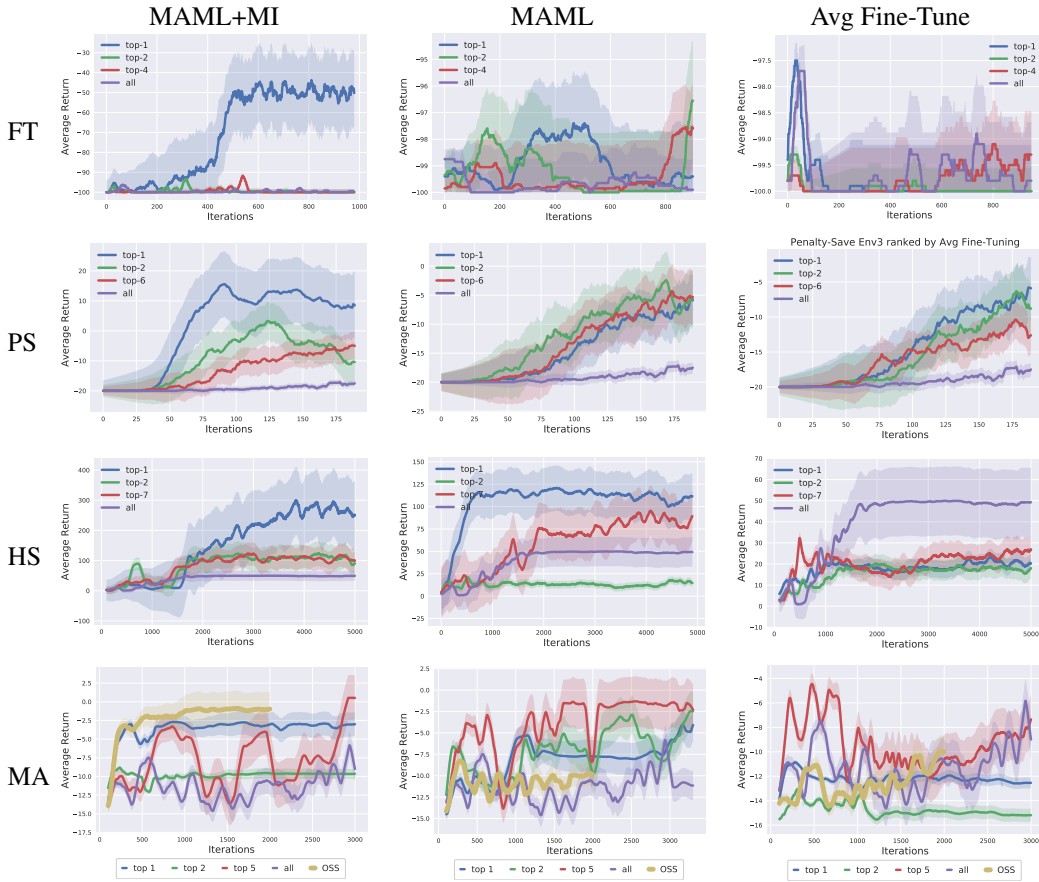

Figure 3: Reward curves for the Top-$K$ demonstration combinations

distance: $\frac{1}{T}\sum_i \left\| s'_{i,t} - \hat{s_{i,t}} \right\| + \left\| a'_{i,t} - \hat{a_{i,t}} \right\|$, where $(\hat{s}_i, \hat{a}_i)$ denotes the trajectories from demonstration $v_i$, and $(s'_i, a'_i)$ denotes the generated trajectories from parameters adapted to $v_i$.

## 5.3 EXPERIMENTAL RESULTS

**Top-$K$ Combinations:** As shown in Fig.3, each row consists of learning reward curves generated from imitating several top-$K$ demonstration subsets composed by each method. For empirical studies, suppose there are $N$ demonstrations in the given noisy set, we select subsets from top-1, top-2, top-$N/2$, and top-$N$, which is simply learning from the entire original noisy set.

- **Free Throw (FT):** as shown in Fig.3 top row, MAML + MI outperforms other baselines by resulting in better composition of top-$K$ demonstration sets and successfully selects the most suitable demonstration from the set. The learning curves for the 20 demonstration set consisting behaviors from a taller expert agent can be found in the appendix C.2.

- **Penalty Save (PS):** as shown in Fig.3 second row, MAML + MI also outperforms other baselines. Notice that since this environment is a multi-goal learning task, the DSA should depend on the environment variants as Eq.9. The curves we show here are trained under a fixed environment variant, which are better illustrated in the videos.

- **Handstand (HS):** as shown in Fig.3 third row, while learning from the top-$K$ subsets selected by MAML + MI achieves the best results, MAML also selects a considerably suitable top-1 demonstration. However, top-2 of MAML performs even worse than the baseline, implying that the second highest ranked demonstration negatively affect the agent outweighing the positiveness of the top-1 ranked demonstration. All of the top-$K$ combinations composed by Avg Fine-Tune perform worse than the baseline supposedly due to wrongly selecting detrimental demonstrations.

- **Martial Arts (MA):** as shown in Fig.3 fourth row, it can be noticed that MAML + MI successfully achieves the task goal using the top-1 demonstration within 2,000 iterations, but neither MAML nor

Avg Fine-Tune succeeded even being trained further. For this environment, we further conducted experiments on generalization to unseen demonstrations and our approximate optimal subset selection algorithm, denoted as *OSS*. The golden thick curves are the learning reward curves imitating the selected subset by the *OSS* for each method. It can be seen that the performances of learning from our selected subset is significantly better in MAML + MI case, while other baselines fail to utilize this algorithm to select better suitable demonstration subset. Learning curves generated adapting to unseen demonstrations can be found in appendix C.3.

**Qualitative Results** The videos showing how the meta-trained parameters can adapt to certain demonstrations and their comparisons with different methods are shown on the project page.

**Behavioral Cloning:** The results on handstand and martial arts can be found in appendix C.4.

**ABD:** Table 1 shows the computed behavioral distances between the adapted behaviors and the expert behaviors, across the entire demonstration set for a mean score. It can be shown that MAML + MI outperforms other baselines by adapting better to the sampled demonstrations. MAML works the best in the free throw environment, potentially due to posterior network $Q$ was not well-trained using the small size of dataset.

| Environment | MAML + MI | MAML | Avg Fine-Tune |
|---|---|---|---|
| Free Throw | 5.07 | **4.99** | 5.39 |
| Penalty Save | **3.03** | 3.11 | 4.63 |
| Handstand | **8.99** | 9.36 | 9.58 |
| Martial Arts | **8.14** | 8.86 | 8.50 |

Table 1: Adapted Behaviors Distance (ABD), the lower the better

## 5.4 EFFICIENCY ANALYSIS

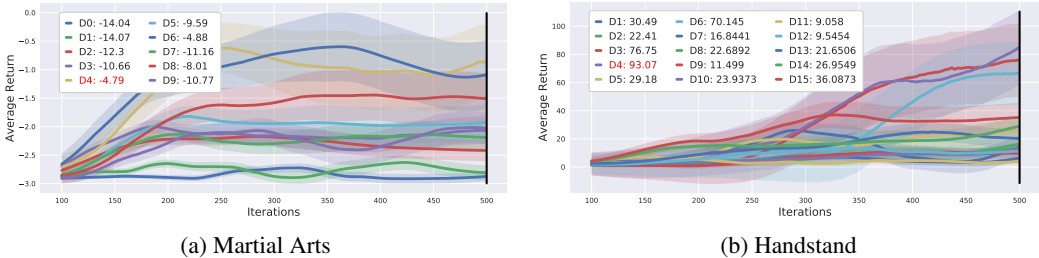

(a) Martial Arts  (b) Handstand

Figure 4: Heuristics Scores for imitation learning from scratch at 500 iterations

In order to verify the efficiency as compared to demonstration judging via training from scratch, We set a limited *quota* be the total number of iterations required for meta-training the adaptive parameters $\theta$ plus the number of gradient updates during adaptation to each demonstration, which sums up to around 6,000 and 7,500 for our martial arts and handstand environments respectively. We then evenly distribute this *quota* to train different agents imitating each of the demonstration from the noisy set, and output their top choices. In Fig.4, the heuristics curves generated up to 500 iterations produce the required assessment scores, we verified that the top choices from both environments are of non-suitable demonstrations, the associated reward curves are presented in the appendix C.6.

## 6 CONCLUSION

We propose a framework to tackle the challenging problem - learning a good policy through imitation learning from a noisy demonstration set. Our framework, built on top of MAML with a mutual information maximized regularization, learns a set of adaptive parameters from the given noisy set. The agent should exhibit significant learning outcomes after fast adaptation to certain demonstrations where these outcomes can be evaluated via predefined task heuristics. By being a learning framework, the system learns to discover the most suitable demonstrations for the agent from the expert rather than selecting based on hand crafted judging rules. For future research direction, we hope this work can serve as the first trial to lure more advanced research on tackling imitation learning from noisy demonstration sets.

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

## A    IMPLEMENTATION AND TRAINING DETAILS

### A.1    TASK HEURISTICS

For evaluating the suitability of certain demonstrations, we require a task dependent knowledge to judge from the imitation learning outcomes after the adaptation to a particular demonstration. We hereby describe the task heuristics for each environment in the following:

- **Free Throw:** The minimum distance between the basketball and the fixed basket.
- **Penalty Save:** The minimum distance between the agent and the incoming shot in an episode.
- **Handstand:** The height of two feet of a humanoid if both two hands are touching the ground without letting the head to hit the ground. We accumulate this heuristics score when the aforementioned body pose condition is satisfied.
- **Martial Arts**: The minimum distance between right foot and the target if the head of the humanoid is higher than a pelvis.

Setups of our simulation environments are listed in Table 2.

| Environment | Observation Space | Action Space | Task Success Reward |
|---|---|---|---|
| Free Throw | 55(continuous) | 55(continuous) | 100.0 |
| Penalty Save | 91(continuous) | 8(continuous) | 20.0 |
| Handstand | 55(continuous) | 21(continuous) | 1000.0 |
| Martial Arts | 55(continuous) | 21(continuous) | 1000.0 |

Table 2: Setups of our simulation environments

### A.2    MODEL ARCHITECTURES

We build the policy networks for each of the simulation environments using a 2-layer MLP. The posterior network $Q$ used in MAML + MI is implemented as a siamese 1-layer Long-Short Term Memory (LSTM) (Hochreiter & Schmidhuber, 1997) network followed by a two consecutive fully-connected networks to produce the 1-dimensional embedding as the prediction of the encoded demonstration feature. In this paper, we simply use l2-norm of observations throughout the entire episode of a demonstration as its encoded feature.

### A.2.1    IMPLEMENTATION DETAILS

For each environment, we used a meta batch size of 10 (except for one of the Free Throw dataset consisting of 8 demonstrations, we used 8) training for up to 2,000 meta updates. We followed the criteria described in the original MAML work (Finn et al., 2017a) for choosing the MAML models with best average return as our adaptive parameters. The learning rate in the inner objective Eq.1 is set to be $\alpha = 1.0$ with KL-divergence constraints introduced by TRPO (Schulman et al., 2015) set to 0.15. $\alpha$ is halved after the first iteration during adaptation for free throw and penalty save environments. For handstand and martial arts environments, we enlarge $\alpha$ by twice for the first iteration to 2.0, and then use 0.5 like the two aforementioned environments. All our resulting curves are plotted with mean and 1 standard deviation of several trials.

Table 3 summarizes the network architecture for the policy networks and the posterior networks $Q$ for each environment.

## B    PSEUDO CODES FOR OUR ALGORITHMS

Algorithm1 summarizes the proposed method for a training the fast adaptive set of parameters. During meta-testing, we adapt the meta-learned parameters to each demonstration in the set one at a time. Algorithm2 summarizes the proposed algorithm for meta-testing as the suitability assessing phase of our framework. We will use the same terminologies as in Sec. 3 and Sec. 4 in the main paper. For

| Environments | Networks | | |
| --- | --- | --- | --- |
| | Policy Network $\theta$ | Posterior Network $Q$ | Discriminator $D$ |
| Free Throw | MLP(32,32) | Siamese-LSTM(128,1) | MLP(100,100,1) |
| Penalty Save | MLP(32,32) | Siamese-LSTM(128,1) | MLP(100,100,1) |
| Handstand | MLP(64,64) | Siamese-LSTM(256,1) | MLP(100,100,1) |
| Martial Arts | MLP(32,32) | Siamese-LSTM(256,1) | MLP(100,100,1) |

Table 3: Network architectures used in this paper, the MLP($h_1$,$h_2$) are two-layer multi-layer perceptrons with hidden sizes $h_1$ and $h_2$. Siamese-LSTM($h_1$, $L$) indicates how $L$ layers of LSTM cells with hidden szie $h_1$. MLP($h_1$,$h_2$,...,$h_n$) for discriminators are multi-layer perceptrons with hidden sizes as indicated.

example, we will denote the horizon of a trajectory as $T$ to differentiate it from the entropy term $H$. Algorithm3 describes the proposed subset selection algorithm.

---

**Algorithm 1** Info Meta Imitation Learning (Blued lines are GAIL ver.)

---

**Require:** $V = \{v_1, v_2, ..., v_n\}$: A set of demonstrations
**Require:** $E = \{e_1, e_2, ..., e_k\}$: An environment with $k$ goals ($k = 1$ for 3 of our environments)
**Require:** $\alpha$, $\beta$: step size hyperparameters
1: randomly initialize $\theta$, $Q$, $D_{meta}$
2: Sample a batch of Demonstrations $v_i \in V$
3: **while** not done **do**
4:    **for all** $v_i$ **do**
5:       Sample a batch of environments $e_j \in E$
6:    **for all** $e_j$ **do**
7:       Sample $K$ trajectories $\tau = \{(s_1, a_1, ..., s_T)\}$ using $f_\theta$ for updating $D_{meta}$ to $D'_{meta,i}$
8:       Evaluate $\nabla \mathcal{L}_{IL_i}(f_\theta)$ using $\tau$ and $v_i$, and update $\theta'_i = \theta - \alpha \nabla \mathcal{L}_{IL_i}(f_\theta)$ with TRPO
9:       Update $D_{meta}$ to $D'_{meta,i}$ with the gradient:
$$\hat{\mathbb{E}}_{\tau_i}[\nabla log(D_{meta}(s, a))] + \hat{\mathbb{E}}_{v_i}[\nabla log(1 - D_{meta}(s, a))]]$$
10:       Sample $K$ trajectories $\tau'_{ij} = \{(s'_1, a'_1, ..., s'_T)\}$ using $f_{\theta'_i}$ after imitating $v_i$ under $e_j$
11:       Update posterior $Q$ with $\tau$, $\tau'_{ij}$, and encoded $v_i$
12:       Sample new trajectories using $f_\theta$ and $f_{\theta'_i}$ as $\tau_{new}$ and $\tau'_{ij,new}$ respectively
13:       Compute $R_{MI_i}$ with $Q$, using newly sampled $\tau_{new}$ and $\tau'_{ij,new}$ as (5)
14:       Evaluate $\nabla \mathcal{L}_{MI_i}(f_\theta)$ with $R_{MI_i}$, update $\theta''_i = \theta'_i - \alpha \nabla \mathcal{L}_{MI_i}(f_{\theta'_i})$ with TRPO as (7)
15:       Sample K trajectories $\tau''_{ij} = \{(s''_1, a''_1, ..., s''_T)\}$ using $f_{\theta''_i}$
16:    **end for**
17:    **end for**
18:   Meta Update the $\theta$ using all the $\tau''_{ij}$ with TRPO update rules as (2)
19:   Meta Update the $D_{meta}$ using all the gradients collected from $D'_{meta,i}$ as (3)
20: **end while**

---

## C MORE RESULTS

### C.1 ADAPTATION CURVES

In Fig.5, we show the adaptation curves from each of the method in every environments we tested on.

### C.2 FREE THROW WITH TALLER AGENT

As shown in Fig.6, both top-1 and top-2 obtained by MAML + MI are suitable demonstration subsets, with demonstration 1 and 5 as top-2 as depicted in Fig.5 top row. Demonstration 2 in this set is actually a demonstration recorded from a much taller agent, which we show in this top-$K$ curves it is not selected as a suitable one despite being a successful demonstration achieving the free throw task. This phenomenon is better illustrated in the videos.

---

**Algorithm 2** Meta-Testing as Judging (Blued lines are GAIL ver.)

---

**Require:** $V = \{v_1, v_2, ..., v_n\}$: A set of demonstrations
**Require:** $E = \{e_1, e_2, ..., e_k\}$: An environment with $k$ goals ($k = 1$ for 3 of our environments)
**Require:** $\alpha, \beta$: step size hyperparameters
**Require:** max_iter: maximum iterations for the meta-testing
**Require:** $\theta$: the meta-trained parameters
**Require:** Pre-defined Task Heuristics $\mathcal{K}$
**Require:** Meta-Trained Discriminator $D_{meta}$
 1: **for all** $e_j \in E$ **do**
 2:    **for all** $v_i \in V$ **do**
 3:       Initialize $\theta_{ij}$ as meta parameters $\theta$
 4:       Initialize $D_i \leftarrow D_{meta}$
 5:       **for** iter = 0 to max_iter **do**
 6:          Sample $K$ trajectories $\tau = \{(s_1, a_1, ..., s_T)\}$ using $f_{\theta_{ij}}$
 7:          Update $D_i$ with the gradient:
                 $\hat{\mathbb{E}}_\tau[\nabla log(D_i(s, a))] + \hat{\mathbb{E}}_{v_i}[\nabla log(1 - D_i(s, a))]]$
 8:          Compute $R_{IL}$ imitating $v_i$ with $D_i$
 9:          Evaluate $\nabla\mathcal{L}_{IL}(f_{\theta_{ij}})$ using $\tau$ and $v_i$, and update $\theta_{ij} \leftarrow \theta_{ij} - \alpha\nabla\mathcal{L}_{IL}(f_{\theta_{ij}})$ with TRPO
10:       **end for**
11:       Evaluate $\theta_{ij}$ for $M$ random seeds
12:       Compute DSA score $S_{v_i}|_{e_j} = \mathcal{K}(\tau_{\pi_{\theta_{ij}}}|_{e_j}) - \mathcal{K}(\tau_{\pi_\theta}|_{e_j})$
13:    **end for**
14:    Record the index $i_{opt}$ for the most suitable demonstration under $e_j$
          with $i_{opt} = \arg\max_i S_{v_i}|_{e_j}, \quad \forall v_i \in V$
15: **end for**

---

**Algorithm 3** Optimal Subset Selection Algorithm

---

**Require:** $V = \{v_1, v_2, ..., v_n\}$: a set of demonstrations
**Require:** $N$: predefined iterations
**Require:** $\beta$: constant for multiplying and dividing weights of demonstrations
**Require:** $\epsilon$: threshold for selecting final subset of demonstrations
**Require:** $W = \{w_1, w_2, ..., w_n\}$: weights for meta-learning each demonstration.
 1: Initialize all $w_i$ to 1
 2: **while** not done **do**
 3:    Update $\theta$ with *Imitation Learning* using current set $V$ for $N$ iterations, applying weights $W$
       for the demonstrations during meta-update.
 4:    Get heuristic score for meta-trained parameters $\theta$, store in $S$
 5:    **for all** $v_i$ in $V$ **do**
 6:       Set $\Delta_{\mathcal{K},ij} = S_{v_i}|_{e_j} - S$
 7:       **if** $\Delta_{\mathcal{K},ij} > 0$ **then**
 8:          Set $w_i = w_i \cdot \beta\exp(\Delta_{\mathcal{K},ij})$
 9:       **else**
10:          Set $w_i = w_i \cdot \beta^{-1}\exp(-\Delta_{\mathcal{K},ij})$
11:       **end if**
12:    **end for**
13:    $S_{prev} = S$
14: **end while**
15: Set $V = \{v_i \mid w_i > \epsilon\}$
16: **return** V.

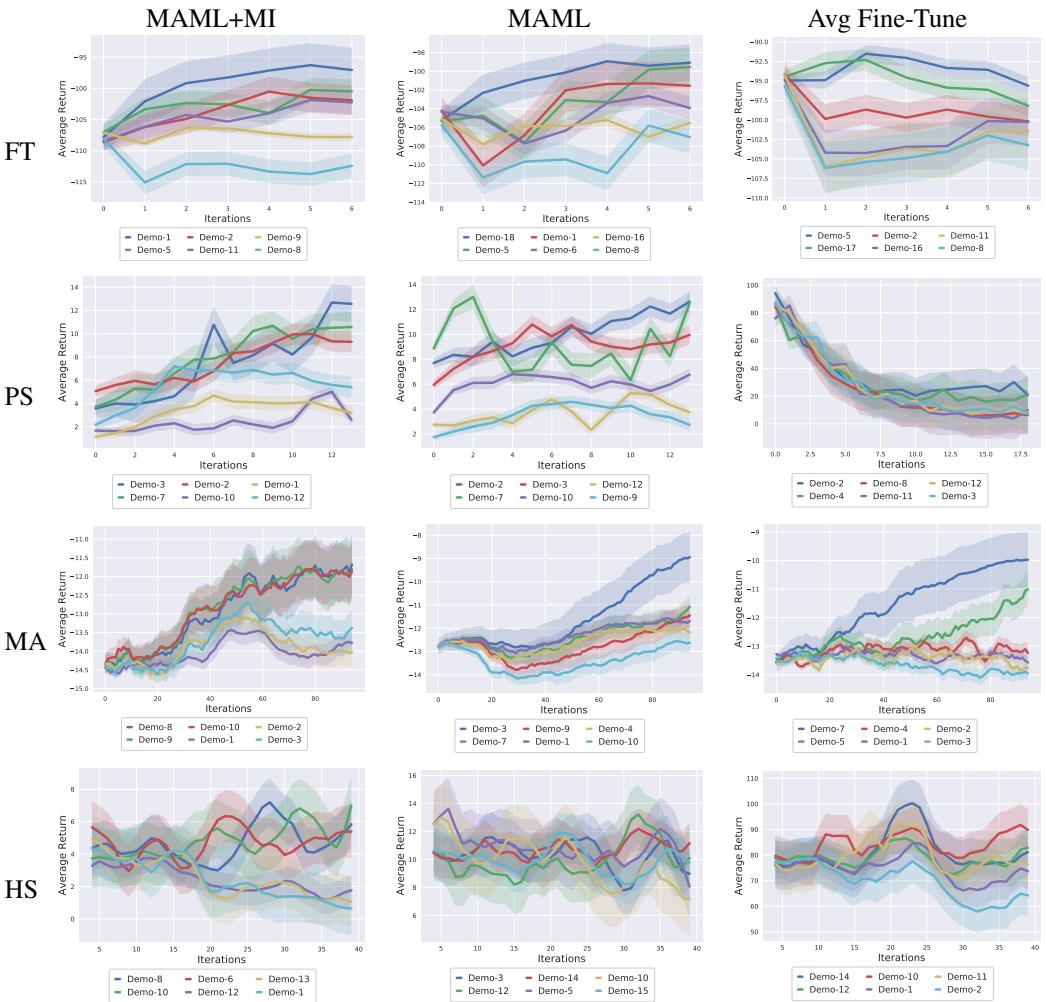

Figure 5: Reward (Heuristics) curves during adaptation for each method

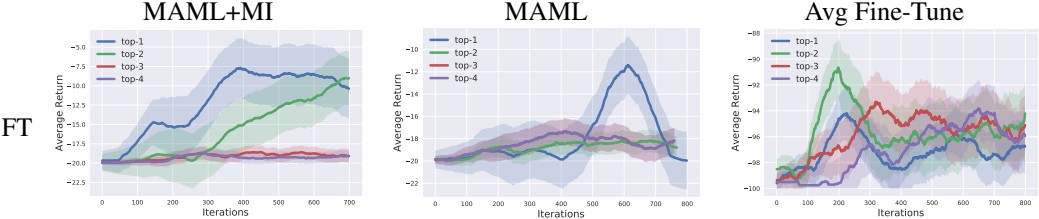

Figure 6: Learning Free Throw from noisy set including demonstrations from non-suitable agent

## C.3 UNSEEN DEMONSTRATION IN MARTIAL ARTS

In Fig.7 we show that our framework can also work on selecting suitable demonstrations given an unseen demonstration set. We meta-imitation-learned on a large set of demonstrations consisting of 100 martial arts demonstrations and tested it on a disjoint random sampled demonstration set. From the reward curves, although requiring more number of iterations to train, MAML + MI selects a subset that eventually succeeds achieving the goal of the task, while other two baselines struggle finding a suitable top-$K$ subset.

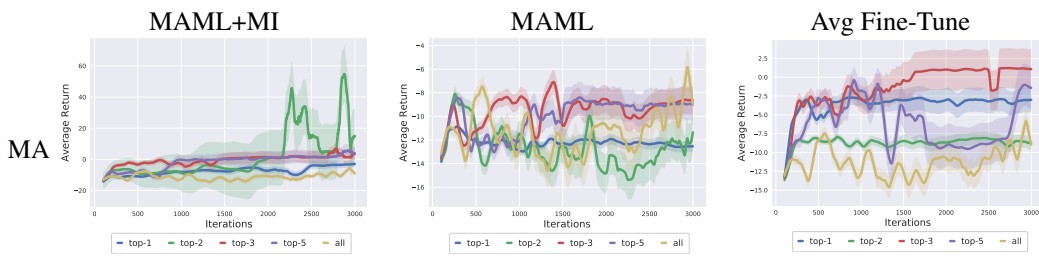

MA

Figure 7: Reward curves for the Top-$K$ unseen demonstration combinations in Martial Arts

## C.4 IMITATION LEARNING VIA BEHAVIORAL CLONING

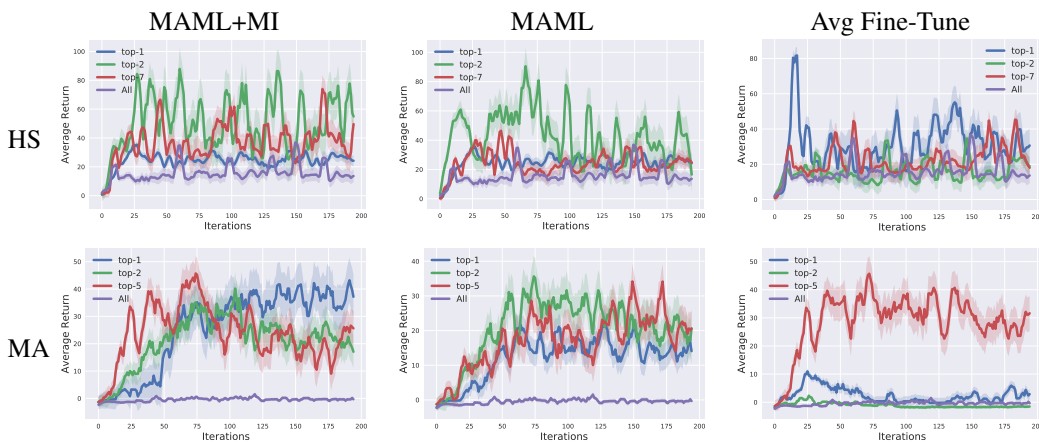

Figure 8: Reward curves for the Top-$K$ demonstration combinations using behavioral cloning as the imitation learning method

In Fig.8, we show learning reward curves using behavioral cloning as the inner imitative update in Eq.1 on two of our harder environments, handstand and martial arts. For handstand environment, the top-2 and top-7 produced by MAML + MI are significantly consistently better than those from other two baselines, implying the top 7 ranked demonstration selected from MAMl + MI are of the more suitable subset. Interestingly, MAML produce reasonably good top-2 as well, which is almost on par with the MAML + MI. We hypothesize that behavioral cloning does not introduce too much randomness requiring further regularizing vanilla MAML. However, behavioral cloning can not produce perfect handstand results due to the complexity of such task, while our primarily adopted imitation learning algorithm GAIL successfully produced perfect imitative outcomes. For martial arts, both MAML + MI and MAML produce reasonable results, while MAML + MI produce better top-1 achieving better overall performances. The Avg Fine-Tune method surprisingly works in this environment with behavioral cloning, hypothetically due to the steadiness of supervised training.

## C.5 SUBSET SELECTION METHOD

The trends of each subset weight $c_i$ and how they change throughout the subset selection training is as shown in Fig.9. We again sub-sample only the best 3 evaluated weights and the worst 3. It is obvious that MAML + MI diverges in a more consistent fashion and produce more reasonable final results.

## C.6 RESULTS FROM EFFICIENT ANALYSIS

In Fig.10, we show the learning reward curves imitating the top-$K$ demonstration combinations generated by training from scratch and snapshot at 500 iterations as described in Sec.5.4. It is obvious that these learning outcomes are significantly worse than those presented in Fig.3 generated

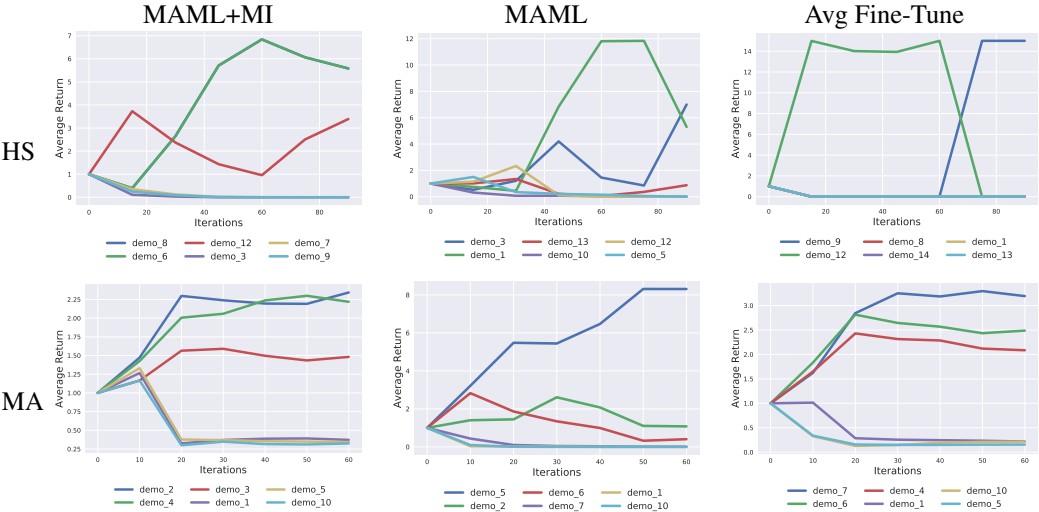

Figure 9: The trends of weights from our subset selection method

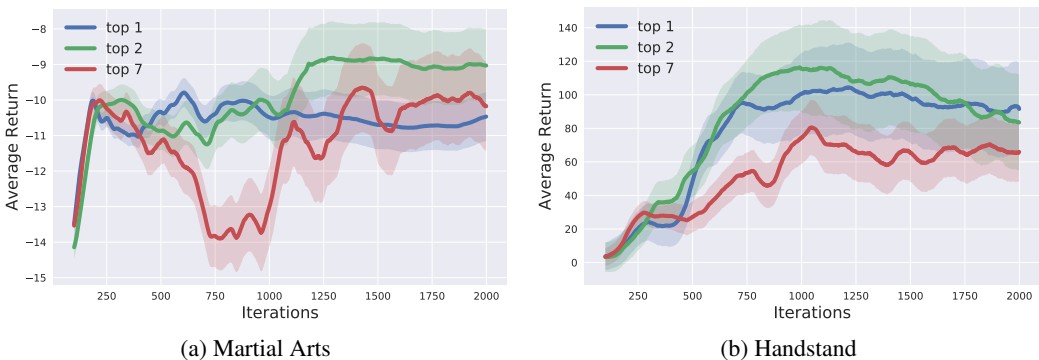

(a) Martial Arts            (b) Handstand

Figure 10: Reward curves for the Top-$K$ demonstration combinations judged by training from scratch

by MAML + MI and MAML. This verifies that in most cases, training from scratch is not only inefficient but also prone to more randomness if reinforcement learning is involved (in our case, GAIL is involved).

