# OpenReview forum: "Learning from Noisy Demonstration Sets via Meta-Learned Suitability Assessor"
_ICLR.cc/2019/Conference_

### Official Review · AnonReviewer2 · 2018-11-03
**many things unclear, experiments not convincing enough, writing needs improvement.**

**Rating:** 4
**Confidence:** 3

**Review:**

The paper makes its intent plainly clear, it wants to remove the assumption that demonstrations are optimal.  Thus it should show that in a case that some demonstrations are bad, it outperforms other methods which assume they are all good. The method proposed, while interesting, well-conceived and potentially novel, is not convincingly tested to this end.

The paper should also show that the method can detect the bad demonstrations, and select the good demonstrations.

The experiments are on toy tasks and not existing tasks in the literature. Why not use an existing dataset/domain and simply noise up the demonstrations?

Furthermore, many crucial details are omitted, such as the nature of the heuristic function K, and how precisely the weighting $c_i$ is adapted (section 4.4). Is it done by gradient descent? We would have to know what K is, and if it is differentiable to know this.

Also the writing itself needs a thorough revision.

I think there may well be promise in the method, but it does not appear ready for publication.

---

### Official Review · AnonReviewer1 · 2018-11-15
**Artificial problem class which doesn't justify the complexity of the method that doesn't deliver good performance.**

**Rating:** 4
**Confidence:** 4

**Review:**

The problem is described as doing imitation learning from a set of demonstrations that includes useless behavior. Authors propose a method that is an extension of MAML which selects the useful demonstrations by their provided performance gains at the meta-training time.

Paper clearly demonstrates significant amount of work. Pieces from different modern method implementations (like MAML, TRPO, GAIL, multiple custom loss functions) are combined to work together. Also four custom task domains are implemented with MuJoCo. Finally decent amount of experiments are run.

Unfortunately, all that hard work can't be justified by the motivations that are very artificial in details and by the final task performance.

First of all, the setup includes small number of demonstrations where almost none of them are seemingly successful (judging by the videos). This is a very artificial setting that does not reflect the actual imitation learning problems like demonstrations provided by humans. There, normally the problem is either dealing with small number of demonstrations that are all typically successful but similarly suboptimal or dealing with small number of distinct demonstrators which are again successful but have significantly different styles. In the summary video, authors motivate the case by learning from sources like internet videos, but that setting is also very far away from the case here, because such video collections are much larger but more importantly the main problem is dealing with the third person perspective. All the experiments here is done from first person demonstrations (in one case with a slightly different body).

Biggest caveat of the paper is that it is promoted as a purely imitation learning method. Yet everything hinges on the existence of a "task heuristic" which is nothing but a reward function. If such function exists, all these first person demonstrations can be judged and selected based on that function. There would be no need for a complicated meta-learning scheme. Also the task could be trained directly on that reward by reinforcement learning. Also computation of this heuristic function is not specified. As far as I understand, it is a different quantity than the sparse "Task Success Reward".

Finally, the final performance of the imitating agents are far from accomplishing the task, though they show some resemblance to the imitation behavior. This is not all that surprising, given small number of demonstrations and high dimensional control problems.

Overall, the details of the setup makes the problem very artificial, the final performance is not impressive. Method is an amalgamation of bunch other recent work, which gives the impression of creating complexity for its own sake. I do not think that this method will be useful for moving the field forward and produce any impact.

---

### Official Review · AnonReviewer4 · 2018-11-16
**Problem of limited scope, with interesting domains but uncompelling final performance**

**Rating:** 4
**Confidence:** 4

**Review:**

Summary/Contributions:
This paper focuses on an imitation learning setup where there some of the provided demonstrations which are irrelevant to the task being considered. The stated contribution of the paper is a MAML based algorithm to imitation learning which automatically determines if the demonstrations are "suitable".  The authors also employ a mutual information based maximization term between the demonstrations and the pre-update and post update trajectories.

Pros:
- The tasks proposed in the problem seem interesting.

Cons:
- The problem statement seems to be of limited scope.
- The use of the task heuristics seems a bit ad-hoc.
- The final policies are unimpressive

Justification for rating:
The major weakness of this paper in my view are that the setup is of somewhat limited scope since receiving irrelevant demonstrations in the form used by the paper would be unnecessarily costly. The domains considered by the paper seem interesting, but the learned policies are not very compelling. I also feel that the MAML baselines + avg finetuning baselines are somewhat limited giving the new domains. I would appreciate for instance a comparison to off-policy learning methods with demonstrations which the authors discuss in the related work (Hester et al. 2017, Nair et al. 2017, Yang et al. 2018). The justification between using mutual information regularization term also does not seem well-motivated and orthogonal to the problem statement. For instance, a diversity of demonstrations should in principle allow for more information between the demonstrations and the induced change.

Other:
The writing and grammar of the paper needs serious revision. There are error throughout the paper starting from the abstract.

---

### Meta-Review · Area_Chair1 · 2018-12-20

**Confidence:** 5
**Recommendation:** Reject

**Metareview:**

The reviewers raised a number of major concerns including the incremental novelty of the proposed (if any), insufficient explanation, and, most importantly, insufficient and inadequate experimental evaluation presented. The authors did not provide any rebuttal. Hence, I cannot suggest this paper for presentation at ICLR.